# Quasi-Distributed Active-Mode-Locking Laser Interrogation with Multiple Partially Reflecting Segment Sensors

**DOI:** 10.3390/s18124128

**Published:** 2018-11-25

**Authors:** Chang Hyun Park, Gyeong Hun Kim, Suck Won Hong, Hwi Don Lee, Chang-Seok Kim

**Affiliations:** 1Department of Cogno-Mechatronics Engineering, Pusan National University, Busan 46241, Korea; ckdgus6819@naver.com (C.H.P.); gh.kim@pusan.ac.kr (G.H.K.); swhong@pusan.ac.kr (S.W.H.); 2Advanced Photonics Research Institute, Gwangju Institute of Science and Technology, Gwangju 34113, Korea

**Keywords:** mode-locked lasers, quasi-distributed sensor, fiber Bragg gratings, Fabry–Perot interferometer

## Abstract

A new type of quasi-distributed sensor system is implemented using an active mode locking (AML) laser cavity with multiple partially reflecting segments. The mode locking frequency of the AML laser is linearly proportional to the overall lasing cavity length. To implement multiple resonators having multiple reflection points installed in a sensing fiber, two types of partial reflectors (PRs) are implemented for an in-line configuration, one with fiber Bragg grating and the other with a fiber Fabry–Perot interferometer. Since the laser has oscillated only when the modulation frequencies for the mode locking frequency match with the corresponding resonator lengths, it is possible to read the multiple partially reflecting segments along the sensing fiber. The difference between two corresponding mode locking frequencies is changing proportionally with the segment length variation between two PRs upon strain application. The segment length change caused by the applied strain can be successfully measured with a linear sensitivity between mode locking frequency and displacement, linearity over 0.99, and spatial position resolution below meter order.

## 1. Introduction

Fiber optic sensors have been extensively studied for several decades as an alternative to existing sensor systems. The optical fiber sensor is excellent in terms of corrosion resistance without electromagnetic interference and operation under high temperature or high pressures environments. It also has a small weight, making it advantageous in broad-band and long-distance measurement applications. A point-based sensor along the optical fiber is one of the most popular choices for optical fiber sensors with various advantages [1,2,3]. For example, a series of discrete fiber Bragg grating (FBG) sensing heads can measure the change of the center reflection wavelength of the FBG to detect an external variation applied to the FBG [4,5,6,7,8,9]. Interferometer sensing heads can also measure interference pattern changes due to external environment changes using an interference structure, such as a Fabry–Perot interferometer (FPI), Michelson interferometer, and Mach–Zehnder interferometer [10,11,12,13,14,15,16,17]. The small length of point-based sensing heads using an FBG sensor or interferometer sensor offers a high position resolution along optical fiber in addition to the advantages of high sensitivity, geometric diversity, and rapid response [18]. The discrete sensing position along the optical fiber is determined by the separated locations of multiple sensing heads. This implies that the optical fiber segments between discrete sensing heads are not used for a sensing component, but only for the simple signal transmission medium. The point-based sensors suffer from an extra cost of each sensing heads and limited number of discrete sensing positions. The fabrication process of sensing heads can be simplified when sensing components are based on optical fiber [11]. 

In order to overcome the limitations of the point-based sensor, including the absence of sensing head zones along the optical fiber, distributed fiber sensors have been proposed to simultaneously measure at thousands of sensing positions using a normal unmodified optical fiber as the sensing component with a low cost. Optical frequency-domain reflectometer (OFDR) and optical time-domain reflectometer (OTDR) sensors can continuously measure an external variation along the optical fiber by measuring the back-scattering of light due to Rayleigh, Brillouin, or Raman phenomena in the optical fiber and partial reflector (PR) [1,2,19,20,21]. Studies on such distributed fiber sensors have been actively conducted on various distributed sensing technologies such as Brillouin optical time domain analysis (BOTDA), Brillouin optical time domain reflectometer (BOTDR) and Rayleigh OFDR. Both BOTDR and BOTDA have sensing resolutions in meters over long distances of several tens kilometers [1,22,23], but Rayleigh OFDR has sensing resolution in millimeters at shorter distances over tens meters [1,24]. These distributed fiber sensors have been recently being studied to continuously increase the measurement distance and sensing resolution. 

In this study, we propose a new type of quasi-distributed sensor system to interrogate the enhanced back-reflected signals from cascaded multiple partially reflecting segment sensors in the active-mode-locking (AML) laser cavity configuration. An FBG or fiber FPI was used as a series of PRs along the optical fiber to enhance the back-reflection signal at specific locations and to suppress the natural back-scattering of Rayleigh light in the normal optical fiber. Unlike conventional point-based sensors, the resonant-wavelength information of the FBG or interference pattern of the fiber FPI are not critical parameters for the utilization of the PR in this system. For example, the available sensing point number of FBG in a single fiber is limited below 20 from the ratio between the total wavelength band of light source and individual wavelength band of each FBG [1,2,3,4,5,6,7,8,9]. Instead, the controllable split ratio between back-reflected and transmitted signal intensities is more important parameter of PR for the sensing point number limit of proposed reflectometry because the limit is determined from the intensity reduction of each additional lasing peak. The precise length tuning of multiple optical fiber segments between these PRs is interrogated based on the mode locking frequency of the AML laser, corresponding to the total cavity length produced by each PR. The AML laser oscillates at the specific modulation frequency that matches to the mode locking frequency of cavity length. It means there can be multiple mode locking frequencies corresponding to the multiple cavity lengths, respectively. Thus, it is possible to read the lengths of multiple partially reflecting segments along the sensing fiber by using the mode locking frequency of the AML laser [25]. By monitoring the AML output intensity with the scanning of the modulation frequency, the external variation applied to the optical fiber segment between the PRs can be measured by the linear shift of the mode locking frequency. Therefore, the measurable length range of the sensing area is determined by the numbers of PRs in the cavity, the fiber segment between the PRs and the free spectral range (FSR). The available sensing number of PRs is determined by the reflectivity of the PR, and the fiber segment between the PRs is affected by the linewidth and FSR of mode locking frequency. For characteristics for the strain sensor, linear responses of the strain were experimentally analyzed using three FBG-type PRs and four fiber FPI-type PRs, respectively. It was successfully demonstrated that the AML fiber laser interrogation has a unique quasi-distributed sensing performance with enough linearity and sensitivity, compared with the conventional point-based sensor interrogation methods for FBG and fiber FPI sensors.

## 2. Theory of AML Laser Interrogation

In order to measure the precise length change of multiple optical fiber segments between PRs, the AML laser principle is applied to the multiple fiber cavity configuration corresponding to multiple mode locking frequencies. As an external modulation frequency applies to the gain element, the AML peak laser output is generated only when the round trip time of light passing through the laser cavity exactly matches (or an integer multiple of) the period of the modulated frequency. At this time, the length change of laser cavity affects the round trip time of light and thus the mode locking frequency is also affected from the change of round trip time of light. This means that the cavity length can be directly detected by measuring the intensity variation of AML laser output. Therefore, the changes in cavity length due to external changes, such as temperature and deformation, can be observed by scanning the modulation frequency.

The multiple partially reflecting segment sensors are formed by the multiple AML laser cavities with cascaded multiple PRs. The optimal mode locking frequency corresponding to each total cavity length can be calculated using:(1)$$f_{m}=\frac{N\times c}{n\times L_{cavity}}\;\left(L_{cavity}=L_{main}+2L_{segment}\right)$$
where *f_m_* is the mode locking frequency, *N* is the order of harmonic mode locking, *L_cavity_* is the total cavity length, *L_main_* is the main cavity length including the ring cavity fiber and sensing fiber part up to PR1, *L_segment_* is the segment length, which is the spacing between adjacent PRs, *c* is the speed of light, and n is the effective group refractive index. According to Equation (1), the change of *L_cavity_* can be directly obtained by measuring the change of *f_m_*. For a smaller total cavity length of the AML laser system, a larger value of *f_m_* can be used to induce a wider variation of *f_m_* for the same total-cavity-length change. The sensing speed is determined by the sweeping rate of the AML laser, which is directly related with the repetition rate of the function generator and data acquisition (DAQ) speed. Depending on the multiple total cavity lengths, the modulation of the total cavity gain can be implemented by the optimal modulation frequency sweeping range of the sinusoidal driving current into the SOA. For a higher mode locking frequency range, an in-line electro-optic filter can be also used for the application of the modulation frequency into the laser cavity including a turned-on SOA with a constant driving current.

Figure 1 shows the schematic of a quasi-distributed AML laser interrogation system using multiple partially reflecting segment sensors. Both FBG-type and fiber FPI-type PRs are employed for an enhanced back-reflected signal at specific separated positions between cascaded optical fiber segment sensors. Light reflected from long optical fiber parts, including cascaded segment sensors through a 3-port circulator, enters the ring cavity consisting of 70% of the 70/30 coupler and the semiconductor optical amplifier (SOA). The 30% port on the 70/30 coupler is used as the laser output port in Figure 1. Stage A and Stage B in Figure 1 change the segment length by applying strain between the PRs. The segment length change of the multiple segment sensors between PRs can be observed by the scanning of the modulation frequency into the AML laser. The combination of a function generator and direct-current (DC) supply is used to sweep the modulation frequency. A periodic driving current into the SOA is optimally controlled with the swept modulation-frequency signal with proper biasing with a bias-tee. During the modulation-frequency sweeping, the intensity of the laser output collected by the photodetector is simultaneously measured to detect the multiple peaks corresponding to the mode-locking conditions. Central control with a computer is performed to synchronize the sweeping period between the modulation frequency and intensity and interrogate for the sensor monitoring. 

## 3. Characterization of the PR

Multiple AML laser cavities along the optical fiber out of the circulator are separated by the cascaded multiple PRs, which are implemented in this study using two types of PRs (FBG and fiber FPI). Conventionally, both FBG and fiber FPI have been extensively utilized as an optical fiber sensing head using their spectral characteristics in the wavelength domain. However, in this study, they are utilized only as wavelength-independent PRs by controlling their split ratio between back-reflected and transmitted signal intensities. In this proposed quasi-distributed interrogation system, the large-length optical fiber segments are affected by external environment changes, instead of point-based PRs between these segments.

### 3.1. FBG-Type PR

In general, the principle of Bragg reflection is the basic mechanism of the back-reflected signal for the FBG sensing head. The center reflection wavelength and its reflectivity can be precisely controlled according to the design parameters of the refractive-index variation in the optical fiber core and creation of a one-dimensional periodic structure using an intense ultraviolet (UV) beam. The center reflection wavelength of the FBG is directly related with the period spacing. When temperature or strain cause an external change of the FBG lattice spacing, this change can be linearly monitored through the center-wavelength change of the reflection spectrum. For cascaded multiple FBGs as a point-based sensor system, the conventional FBG interrogation methods based on the wavelength-domain information required a strong, narrow, and non-overlapped reflection spectrum of the FBG [9]. This implies that the strong and narrow reflection of the FBG requires a high power and long time for the UV beam, and the non-overlapped reflection of the FBG spectra requires a complex fabrication process with differentiated periodic refractive index structures. 

However, for the proposed AML laser interrogation system, the FBG does not work as a point-based sensing head, but rather serves as just a point PR for quasi-distributed large-length optical fiber segment sensors. This implies that an FBG having a weak, broad, and overlapped reflection spectrum can be useful as a PR successfully if it provides a relatively higher back-reflected signal from a specific location compared to the Rayleigh back-scattering signal from elsewhere. This identical-wavelength and weak-reflection FBG array has a few advantages including a lower power and short time of UV fabrication, fixed grating period, and low manufacturing cost. It is also expected to produce identical-wavelength and weak-reflection FBGs more easily during the drawing process of optical fiber considering the UV pulse energy and drawing speed. 

We employed identical-wavelength and weak-reflection FBGs, which has a center reflection wavelength of 1312.5 nm and reflectivity of 5.6%. As shown in Figure 2 with the reflection and transmission spectra, the transmission loss of the FBG used in the experiment was measured to be about 0.3 dB. The resulting small transmission loss at 1312.5 nm is helpful to cascade the series of many identical FBGs to increase the measurable length range along the sensing fiber part. Therefore, the FBG PRs in this AML laser interrogation system could overcome the limitations on the available number of cascaded sensing parts and measurable distance of conventional FBG sensor system.

### 3.2. Fiber FPI-Type PR

The PR for the AML laser interrogation system can be also implemented with a fiber FPI structure as it is required to induce a controllable split ratio between back-reflected and transmitted signal intensities without considering the wavelength spectral information. Conventional point-based FPI sensing heads have measured the shift of the interference fringe pattern of the inner FPI cavity due to the external change. Therefore, wavelength-domain interrogation is required for the monitoring of the interference change, and a highly sensitive receiver is required to detect the small back-reflected signal. In this study, a fiber FPI-type PR could be used to easily detect the location change of the back-reflected signal, instead of the challenging detection of the interference fringe pattern change.

Figure 3 shows a schematic of the fiber FPI structure. The ends of the two optical fibers were cut at 90°; *ΔS* is the spacing between the two cutting surfaces, and the cutting surface is fixed to the tube at various *ΔS*. On each cut plane, reflections *R_1_* and *R_2_* occur owing to the refractive-index difference between the glass and air. As *R_1_* and *R_2_* interfere with each other owing to the path difference of the inner FPI cavity, interference fringes are generated depending on *ΔS* for both back-reflected and transmitted signal spectra.

Figure 4 shows measurement results of interference fringes depending on ΔS for both reflection and transmission spectra. The interference pattern is dependent on the increment of the interval *ΔS*. The conventional wavelength-domain interrogation method measures interferograms to measure changes in *ΔS* due to the external environment. 

Therefore, in this study, we focused on the intensity change of the reflected light according to the increment of *ΔS*, instead of counting the number of interference pattern fringes in the wavelength domain. The intensity of the reflected signal is directly affected by *ΔS*; it can be described by Equation (2) [10]. The total reflected signal in the fiber FPI PR is a superposition of two reflected signals, expressed by Equation (2):(2)$$I=A^{2}\left[1+\left(\frac{2ta}{a+2\Delta S\tan(\sin^{-1}(NA))}\right)\cos\left(\frac{4\pi \Delta S}{\lambda }\right)+{\left(\frac{ta}{a+2\Delta S\tan(\sin^{-1}(NA))}\right)}^{2}\right]$$
where *a* is the fiber core radius, *t* is the transmission coefficient of air–glass of approximately 0.98, *ΔS* is the spacing between fibers, and *NA* is the numerical aperture of the fiber. 

If the constants are substituted in Equation (2), the intensity of the reflectance corresponding to a specific *ΔS* can be calculated for each condition.

Figure 5a–e show FPI reflection spectrum according to *ΔS* and Figure 5f Graph of the relationship between *ΔS* and wavenumber variation *Δk.* The *ΔS* was measured while varying the range of 20–100 μm (one step at 20 μm). In the graph of Figure 5f, the linearity of *ΔS* and *Δk* is more than 0.999.

Figure 6 shows experimental measurements of intensity variation of reflected and transmitted light according to *ΔS* of optical fiber FPI using the amplified spontaneous emission (ASE) of SOA with the center wavelength of 1310 nm. Both transmitted and reflected light intensities are affected to oscillate to maintain the mutual intensity total summation upon the change of *ΔS*. When *ΔS* becomes larger than a certain distance, the transmitted light intensity rapidly decreases; however, the reflected light intensity converges to a certain value for a constant reflected signal from the *R_1_* surface. Figure 6 shows that the reflected light intensity is distributed in the range of 0–13%, which implies that a desired reflected signal can be inexpensively implemented by precisely changing *ΔS* with normal optical fibers. We fabricated an FPI structure with reflectivity between 7–9% and used it in the experiment. 

## 4. Interrogation of the Multiple Segment Sensor

### 4.1. FBG-Type Partially Reflecting Segment Sensors

Two types of PRs were used to construct the AML laser interrogation system to demonstrate the feasibility of the quasi-distributed large-length optical fiber segment sensors. First, three FBG-type PRs were used to obtain two segments of quasi-distributed optical fiber sensors between these three positions. As shown in Figure 1, the AML laser cavity oscillates at the mode locking frequency corresponding to each FBG-type PR (PR1, PR2, and PR3). This implies that two segments, PR1–PR2 and PR2–PR3, can be used to obtain an external strain force in the distribution. The monitoring of the peak intensity time during synchronized and repeated scanning of the modulation frequency period is a simple method to reveal the position changes of PR1, PR2, and PR3 and changes of the PR1–PR2 and PR2–PR3 segment lengths.

Figure 7 shows the experimental results for the peak intensity monitoring by sweeping the modulation frequency when an external strain is applied to the PR2–PR3 segment (Stage B in Figure 1). Since the main cavity length, including the ring cavity and sensing part up to PR1, is 16.657 m up to PR1, it corresponds to a mode locking frequency of PR1 is 12.2712 MHz. The mode locking frequencies of PR2 and PR3 are also measured to be 9.8751 and 6.8128 MHz, respectively. It means that the segment length between PR1 and PR2 is 2.021 m, and that between PR2 and PR3 is 4.652 m. Two fiber holders are attached on the sensing piece with a length of 0.75 m in the center region of the PR2–PR3 segment; this sensing piece is stretched from 0 to 1.4 mm with step increments of 0.2 mm. The stretched sensing piece length of 1.4 mm corresponds to the applied strain of 1867 µε. Figure 7a shows the AML intensity distribution of initial mode locking frequencies before applying strain. Each peak time for the corresponding mode locking frequency matches well with the numerical expectation based on the total cavity length. Figure 7b,c show a dB scale spectrum along modulation frequency domain to show signal to noise ratio (SNR) and a linear scale spectrum along cavity length domain to show the distance linewidth, respectively. Most of three peaks have a high SNRs of 54 dB or more and narrow distance linewidths of 13.381, 10.149 and 4.604 cm, corresponding to PR1, PR2 and PR3 peaks, respectively. Because the distance linewidth determines the minimum spacing of the PR segments to distinguish the neighbor peaks during sweeping the modulation frequency, the minimum spacing distance of the PR segments corresponds to the spatial resolution of the given quasi distributed sensor. Therefore, there is requirement that the minimum distance between the PR segments of the experiment set up is 11.765 cm for the PR1–PR2 segment and 7.377 cm for the PR2–PR3 segment, respectively. 

Figure 7d,e show the modulation frequency spectrum of PR1 and PR2, respectively, which did not change within a negligible measurement error as the length of the PR1–PR2 segment did not change. Figure 7f shows the spectral change of PR3 along modulation frequency domain when the segment length increases from 0 to 1.4 mm.

The relationship between the PR3 mode locking frequency shift and PR2–PR3 segment length increment is linearly shown in Figure 8. The sensitivity of the PR3 mode locking frequency shift with respect to the PR2–PR3 segment length increment is 594 Hz/mm, with a linearity R^2^ value of 0.9946. Based on the distance between fiber holders in the center region of the segment of 0.75 m, this sensitivity of length change is corresponding to the sensitivity of applied strain with 445.5 Hz/mϵ.

Figure 9a,b show the length measured stability of PR1 and PR2. Since the positions of PR1 and PR2 and the cavity length between them did not change during the measurement experiment for 160 minutes, the results in Figure 9 show the stability of the sensor system and the corresponding minimum measurable length. The standard deviation of PR1 was measured to about 4.4 μm, and that of PR2 was about 4.5 μm. It means that the minimum measurable length change of the sensor system is limited to about 4.5 μm. Since a SNR of three is generally accepted for estimating limit of detection (LOD) and SNR of ten is used for estimating and limit of quantification (LOQ) [26], it can be expected that the minimum measurable length change of the sensor system can be limited to more than about 45 μm.

Figure 10 shows the peak intensity monitoring along the modulation frequency when the length of the PR1–PR2 segment (Stage A in Figure 1) is changed, instead of changing the PR2–PR3 segment length (Figure 7). Unlike to the case of Figure 7 and Figure 8, the applied strain on the PR1–PR2 segment affects not only the total cavity length of PR2 but also that of PR3, both mode locking frequencies of PR2 and PR3 are simultaneously affected by the increase of the PR1–PR2 segment length. When the main cavity length of PR1 is 17.003 m, its corresponding mode locking frequency is measured to be 12.0215 MHz. As the segment lengths between PR1 and PR2, and PR2 and PR3 are 4.652 m and 2.021 m, respectively, the mode locking frequencies of PR2 and PR3 are 7.7698 MHz and 6.7348 MHz, respectively.

Figure 10a shows the modulation frequency spectrum before applying strain. Figure 10b,c show a dB scale spectrum along modulation frequency domain to show SNR and a linear scale spectrum along cavity length domain to show the distance linewidth, respectively. Most of three peaks have a high SNRs of 54 dB or more and narrow distance linewidths of 12.848, 6.013, and 4.352 cm, corresponding to PR1, PR2 and PR3 peaks, respectively. Therefore, the spatial resolution of the PR1-PR2 segment is 9.431 cm and the spatial resolution of the PR2-PR3 segment is 5.1825 cm. The initial length of the sensing length in the PR1–PR2 segment was 0.75 m; the sensing length was stretched from 0 to 1.4 mm at increments of 0.2 mm, similar to the previous experiment in Figure 7 and Figure 8. It is noted that the total cavity length increased by twice the physical segment length change.

Figure 10d shows the change of the PR1 mode locking frequency, revealing that no considerable change in the mode locking frequency was observed, with a negligible measurement error. Figure 10e,f show the shifts of the mode locking frequencies upon the total cavity length changes of PR2 and PR3, respectively. With the increase of the PR1–PR2 segment length, the total cavity length of PR2 and PR3 located after the PR1–PR2 segment changes together and the corresponding mode locking frequency changes due to the total cavity length.

Figure 11 shows the relationships between the PR2 and PR3 mode locking frequency shifts and strain, obtained upon the increase of the PR1–PR2 segment length. The sensitivity of the PR2 mode locking frequency shift to the PR1–PR2 segment length increment is 461 Hz/mm; the sensitivity with respect to PR3 is 594 Hz/mm. The R^2^ values for both graphs are larger than 0.999. Based on the distance between fiber holders in the center region of the segment of 0.75 m, this sensitivity of length change is corresponding to the sensitivity of applied strain with 345.75 Hz/mϵ.

Figure 11a,b show the both mode locking frequencies of PR2 and PR3 changes together with the single increment of PR1-PR2 segment length. Since the variation of the mode locking frequency of PR2 and PR3 is the same in the graphs of Figure 11a,b, it can be interpreted that the PR2-PR3 segment length was not changed and PR3 was not affected. 

Figure 12 shows the length measurement stability of PR1 when the position of PR1 did not change for 160 min and the stability of the sensor system was measured to have a standard deviation of about 10.2 μm, which corresponds to the minimum measurable length change of more than about 102 μm from the LOQ of the ten times of standard deviation.

In this research, the sensing positions were experimentally implemented using three FBGs of PRs. To calculate the limit of sensing point numbers, the intensity reduction of each additional lasing peak is monitored and it is measured to about 0.3 dB per added FBG-PR as shown in Figure 7b and Figure 10b. It means that more than 80 FBG-PR can be cascaded in the single sensing port for multiple reflection points assuming the undistinguished SNR level of multiple lasing spectra of 48 dB. The minimum distance of the PR segment can be determined based on the distance linewidth from the modulation frequency peak since the modulation frequency peaks of the PR segments must be distinguished without overlapping each other. In addition, since the sensor system measures the change in the length of the PR segment, the minimum distance of the PR segment can be used to determine the spatial position resolution along optical fiber.

### 4.2. Fiber FPI-Type Partially Reflecting Segment Sensors

The quasi-distributed large-length optical fiber segment sensor was also constructed using the fiber FPI as PRs in the Figure 1. To show a potential and characteristics of the additional multiple segment sensing, four fiber-FPI-type PRs were used to measure the characteristics of the three segment sensors. The main cavity length of the ring cavity fiber and sensing fiber part up to PR1 is 26.001 m, which corresponds to a mode locking frequency of 7.8612 MHz. As the PR1–PR2 segment length is 2 m, the PR2–PR3 segment length is 3.424 m, PR3–PR4 segment length is 1.348 m, it can be simply measured that the next mode locking frequency for PR2 is 6.813 MHz, that of PR3 is 5.5468 MHz, and that of PR4 is 5.1688 MHz. The mode locking frequency shift was measured when a 0.55-m-long sensing length in the middle of PR2–PR3 segment (Stage B in Figure 1) was stretched up to 1.0 mm with increments of 0.2 mm. Figure 13 shows the variation of the mode locking frequency according to the initial mode locking frequencies from PR1 to PR4 caused by the single change of the PR2–PR3 segment length only.

Figure 13a shows the mode locking frequency distribution of the fiber FPI-type PRs before applying strain. Figure 13b,c show a dB scale spectrum along modulation frequency domain to show SNR and a linear scale spectrum along cavity length domain to show the distance linewidth, respectively. Most of three peaks have a high SNRs of 25 dB or more and narrow distance linewidths of 38.659, 38.655, 41.254 and 40.465 cm, corresponding to PR1, PR2, PR3 and PR4 peaks, respectively. Therefore, the spatial resolutions of PR1–PR2 segment, PR2–PR3 segment and PR3-PR4 segment are 38.657, 39.955 and 40.86 cm, respectively. Figure 13d,e show that the mode locking frequencies of PR1 and PR2 are unchanged, i.e., unaffected by the change of the PR2–PR3 segment length. Figure 13f,g show the variations of the mode locking frequencies of the PR3 and PR4, installed the far location after the PR2–PR3 segment, respectively. The mode locking frequencies of PR3 and PR4 vary with the length of the PR2–PR3 segment, as the segment length affects the total cavity length according to the mode locking frequency formula. 

Figure 14 shows the correlations between the mode locking frequencies of PR3 and PR4 and the change of PR2–PR3 segment length. 

The strain sensitivities of the PR3 and PR4 mode locking frequencies were 330 Hz/mm and 315 Hz/mm, while the R^2^ values corresponding to very linear relationships were 0.9955 and 0.9957, respectively. Based on the distance between fiber holders in the center region of the segment of 0.55 m, this sensitivity of length change is corresponding to the sensitivity of applied strain with 173.25 Hz/mϵ. Since the graphs of Figure 14a,b shows that the linear change of PR3 and PR4 are simultaneously shifted, it means that PR3 and PR4 were affected by single change origin of PR2–PR3 segment length only, not from that of PR3–PR4 segment length. 

Figure 15a,b show the length measured stability of PR1 and PR2. Since position of PR1 and PR2 did not change for 120 min, the results in Figure 15 show the stability of the sensor system and the corresponding minimum measurable length. The standard deviations of PR1 and PR2 were about 10.6 μm and 9.2 μm, respectively. Thus, the minimum measurable length change of the sensor system is estimated to more than about 106 μm based on the LOQ with the SNR of ten.

## 5. Conclusions

We have demonstrated a quasi-distributed sensor system is implemented using an AML cavity with multiple partially reflecting segments. To implement multiple resonators having a multiple reflection points installed in a sensing fiber, two types of PRs of FBG and FPI are implemented for an in-line configuration. Most of PRs have a good linearity of R2 higher than about 0.99 and a mode locking frequency sensitivity to displacement of 330 Hz/mm or higher, a signal to noise ratio of 24 dB or higher and a spatial resolution of 40.86 cm or less. Owing to the principle of total-cavity-length detection of the AML laser, PRs with small reflectivity can be useful to differentiate the border of multiple segments and distinguish the sequence of each segment. We demonstrated that it can overcome the limitations on the available number of sensing positions and wavelength-domain information of ordinary FBG and interferometer sensing heads. The quasi-distributed sensor system proposed in this study will be useful various applications, such as structure health monitoring and multiple location strain sensing, to combine the advantages of a high sensitivity of the conventional point sensor and wide measurement range of the distributed sensor.

## Figures and Tables

**Figure 1 sensors-18-04128-f001:**
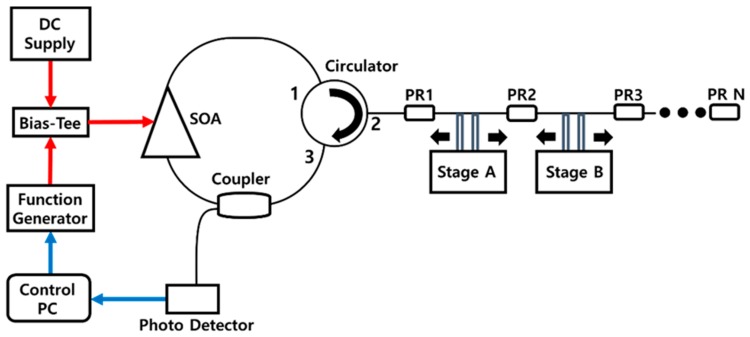
Schematic of a quasi-distributed AML laser interrogation system using multiple partially reflecting segment sensors.

**Figure 2 sensors-18-04128-f002:**
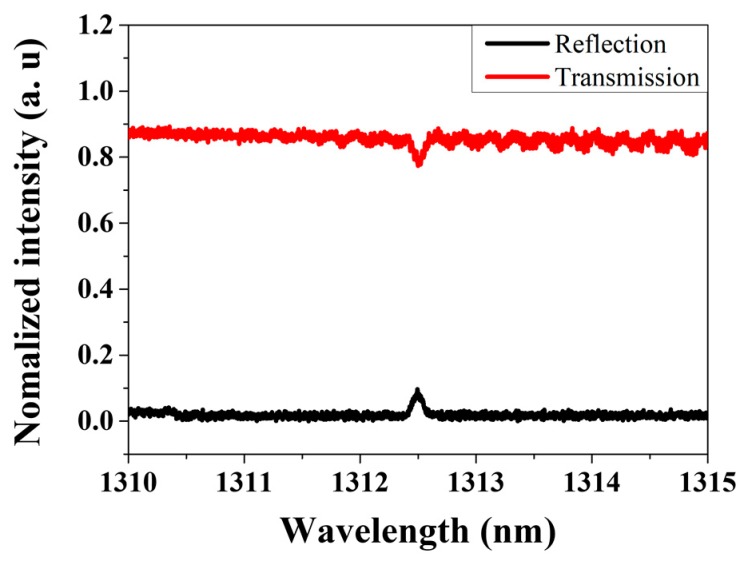
Reflection and transmission spectra of FBG.

**Figure 3 sensors-18-04128-f003:**
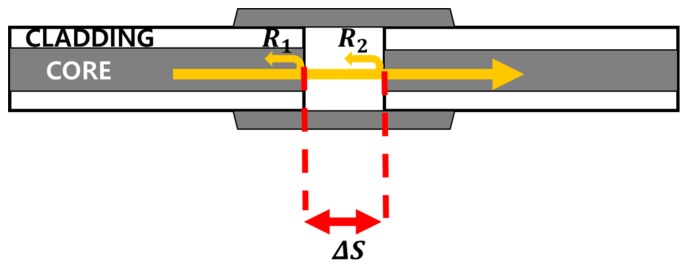
Schematic of the fiber FPI structure.

**Figure 4 sensors-18-04128-f004:**
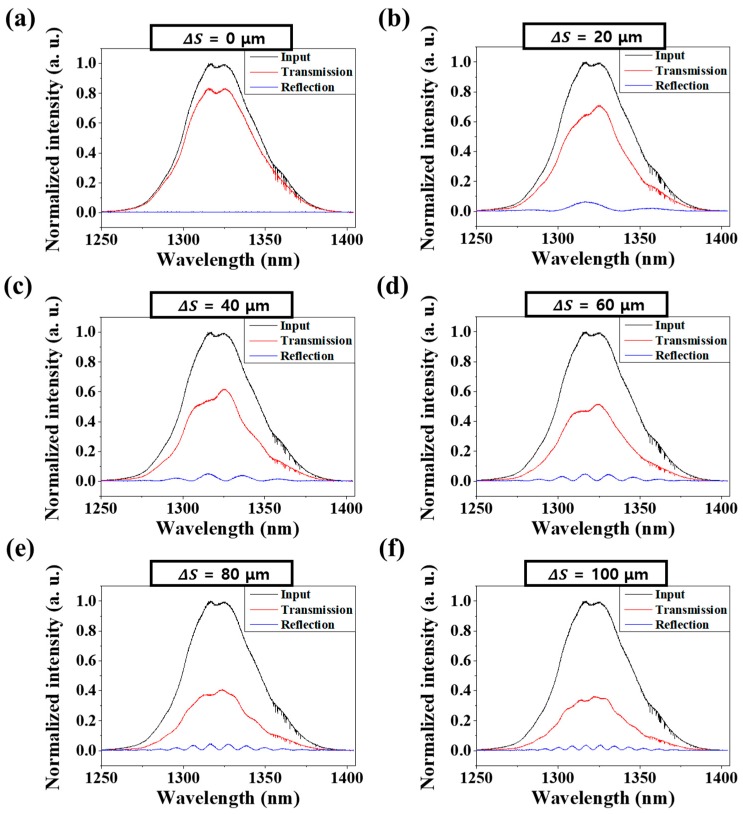
Measurement results of interference fringes depending on *Δ*S for both reflection and transmission spectra.

**Figure 5 sensors-18-04128-f005:**
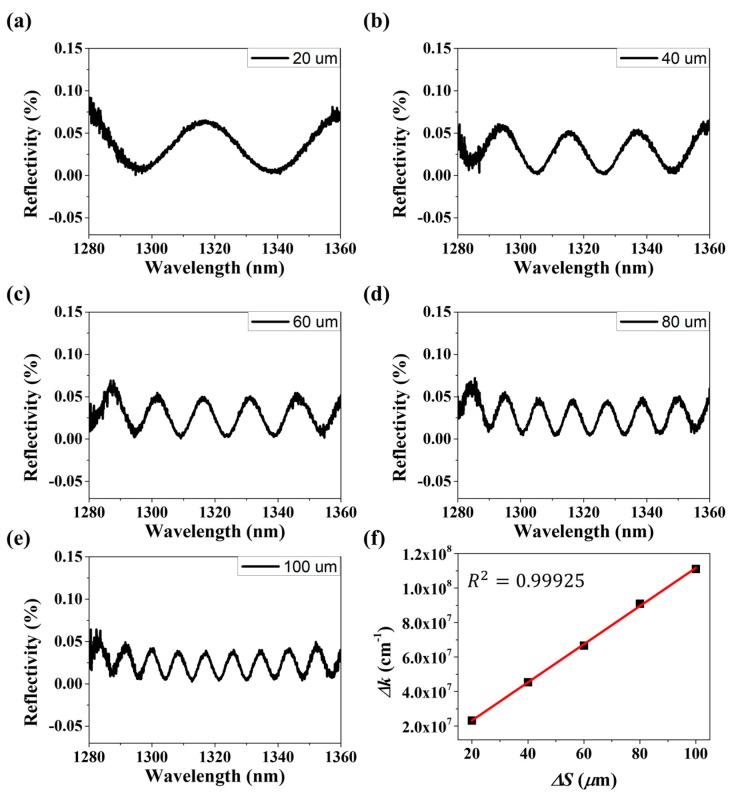
(**a**–**e**) FPI reflection spectrum according to *ΔS*, (**f**) Relationship between *Δ*S and wavenumber variation *Δk.*

**Figure 6 sensors-18-04128-f006:**
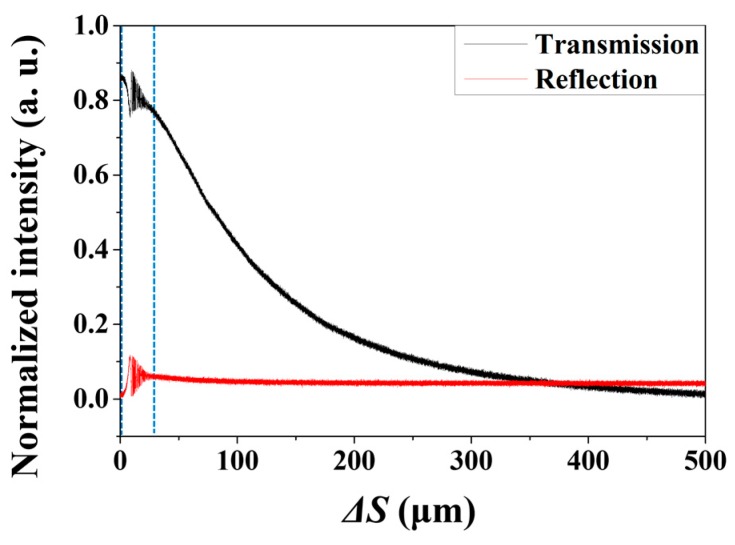
Experimental measurement of intensity variation of reflected and transmitted light according to *Δ*S of optical fiber FPI using the amplified spontaneous emission (ASE) of SOA with the center wavelength of 1310 nm.

**Figure 7 sensors-18-04128-f007:**
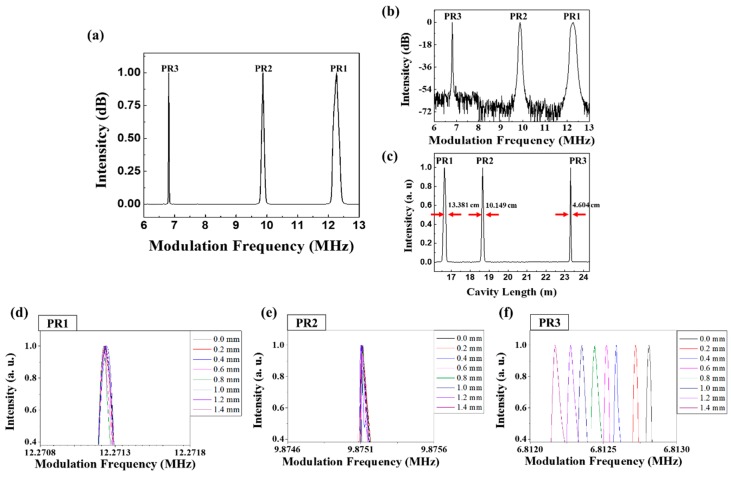
(**a**) The modulation frequency spectrum of PR1, PR2, and PR3 at the initial stage before the strain applies. (**b**) dB scale spectrum along modulation frequency domain to show signal to noise ratio. (**c**) linear scale spectrum along cavity length domain to show the distance linewidth. (**d**–**f**) modulation frequency spectra of PR1, PR2, and PR3, respectively, according to the change of the PR2–PR3 segment length (stage B in Figure 1).

**Figure 8 sensors-18-04128-f008:**
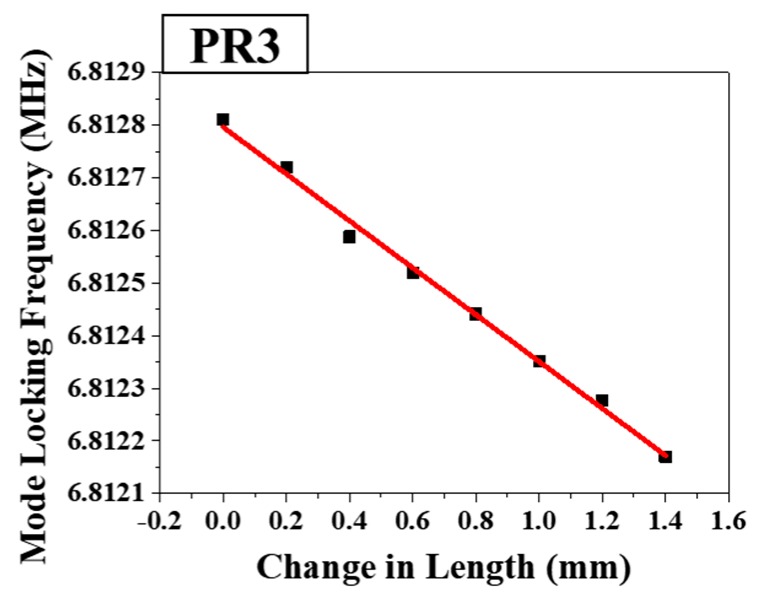
Linear relationship between the mode locking frequency of PR3 and the change in length of the PR2–PR3 segment (stage B in Figure 1).

**Figure 9 sensors-18-04128-f009:**
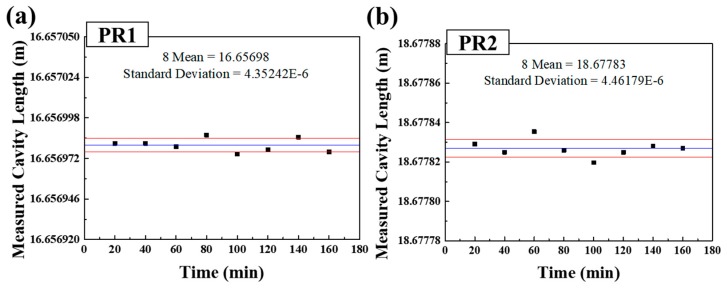
Length measurement stability results of (**a**) PR1 and (**b**) PR2.

**Figure 10 sensors-18-04128-f010:**
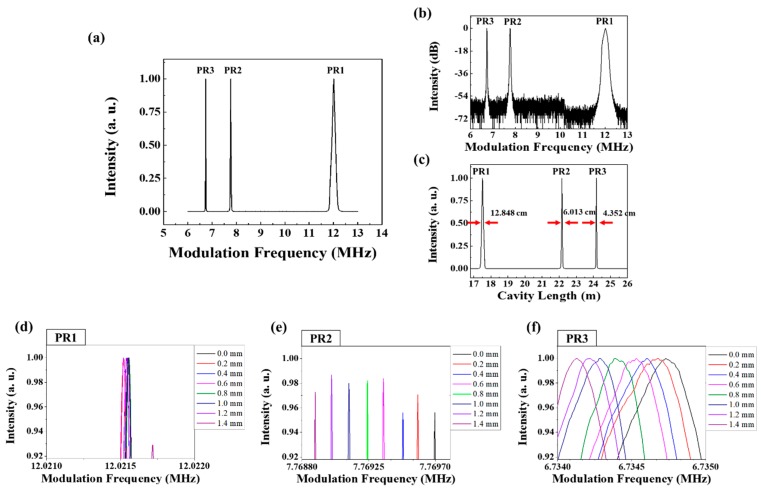
(**a**) The modulation frequency spectrum of PR1, PR2, and PR3 before the strain applies. (**b**) dB scale spectrum along modulation frequency domain to show signal to noise ratio. (**c**) Linear scale spectrum along cavity length domain to show the distance linewidth. (**d**–**f**) Changes of the modulation frequency spectra of PR1, PR2, and PR3, respectively, according to the change of the PR1–PR2 segment length (stage A in Figure 1).

**Figure 11 sensors-18-04128-f011:**
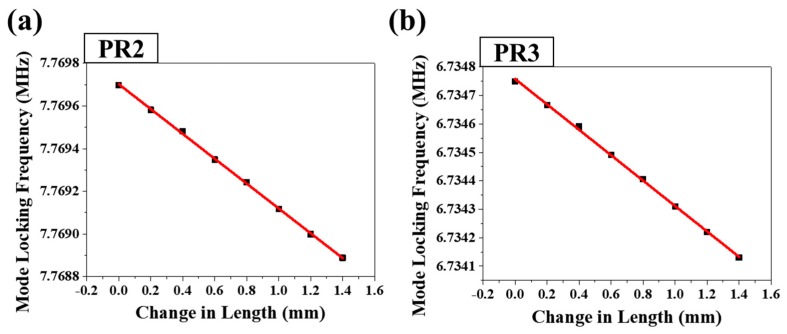
Linear relationships between the mode locking frequencies of (**a**) PR2 and (**b**) PR3 and the change in length of the PR1–PR2 segment (stage A in Figure 1).

**Figure 12 sensors-18-04128-f012:**
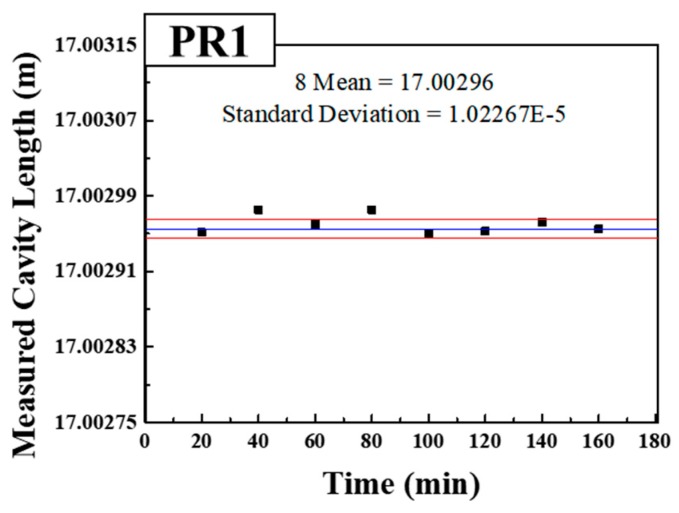
Length measurement stability results of PR1.

**Figure 13 sensors-18-04128-f013:**
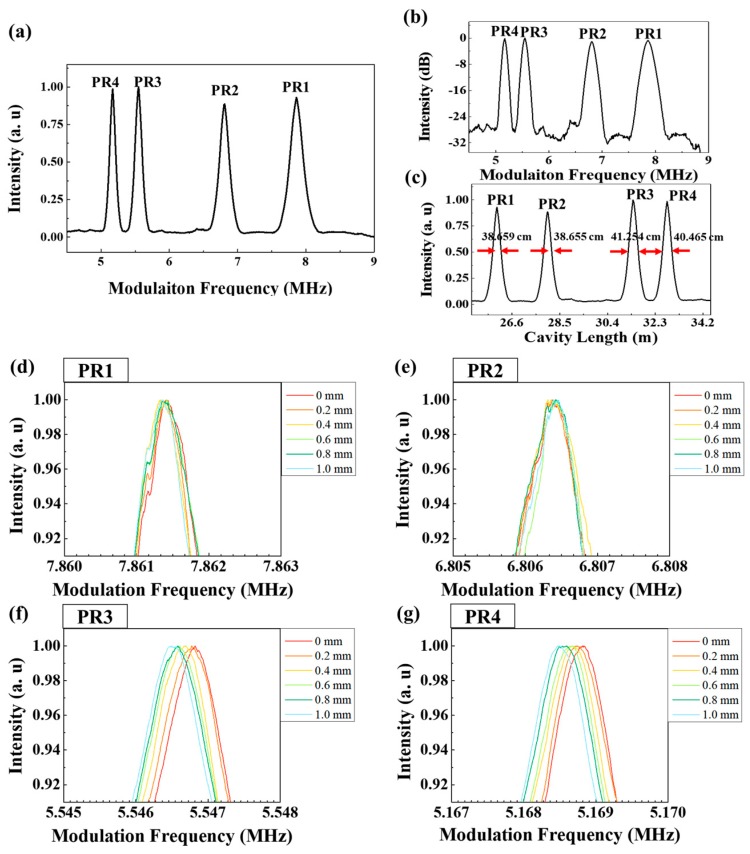
(**a**) The modulation frequency spectrum of PR1, PR2, PR3 and PR4 (**b**) dB scale spectrum along modulation frequency domain to show signal to noise ratio. (**c**) linear scale spectrum along cavity length domain to show the distance linewidth. (**d**–**g**) Modulation frequency spectra of PR1–PR4, respectively, according to the change of the PR2–PR3 segment length (stage B in Figure 1).

**Figure 14 sensors-18-04128-f014:**
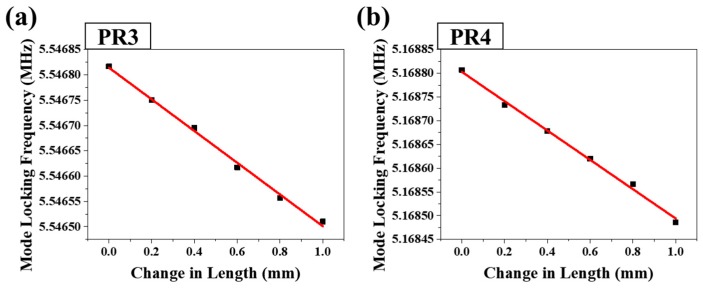
Relationships between the mode locking frequencies of (**a**) PR3 and (**b**) PR4 and the change in length of the PR2–PR3 segment (stage B in Figure 1).

**Figure 15 sensors-18-04128-f015:**
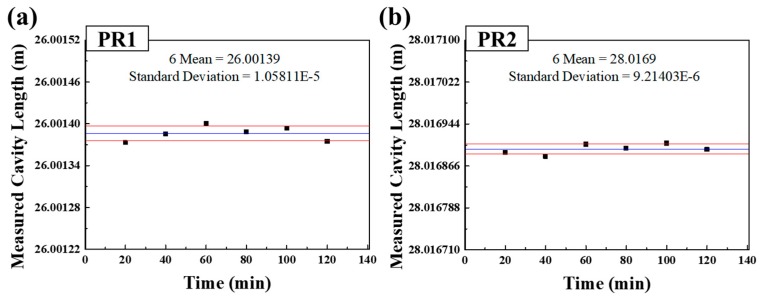
Length measurement stability results of (**a**) PR1 and (**b**) PR2.

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
