# Peer review of "Quasi-Distributed Active-Mode-Locking Laser Interrogation with Multiple Partially Reflecting Segment Sensors"

_sensors, 2018, doi:10.3390/s18124128_

Reviewer 1 Report

The paper is interesting and illustrates experimental results. I suggest shortening the introduction and to add a table for comparing other sensors/sensing systems taken from the literature or available on market from the point e.g of the  sensitivity

Author Response

Point 1: The paper is interesting and illustrates experimental results. I suggest shortening the introduction and to add a table for comparing other sensors/sensing systems taken from the literature or available on market from the point e.g of the sensitivity.

Response 1: Thank you for kind comment on the required comparisons with other sensors/sensing systems. Since the reference 1 is a representative review paper on various sensors such as FBG, BOTDA, and BOTDR mentioned in this paper and there is already a total comparison table of sensors in Reference 1, we summarized the comparison results of this table in our introduction part and explained them with the additional quantitative comparison for those sensors additionally.

Line 57-63 : Studies on such distributed fiber sensors have been actively conducted on various distributed sensing technologies such as Brillouin optical time domain analysis (BOTDA), Brillouin optical time domain reflectometer (BOTDR) and Rayleigh OFDR. Both BOTDR and BOTDA have sensing resolutions in meters over long distances of several tens kilometers [1, 22, 23], but Rayleigh OFDR has sensing resolution in millimeters at shorter distances over tens meters [1, 24]. These distributed fiber sensors have been recently being studied to continuously increase the measurement distance and sensing resolution.

Reviewer 2 Report

In this paper, the authors introduced a new type of quasi-distributed sensor system is implemented using an active-mode-locking laser cavity with multiple partially reflecting segments. The modulation frequency of the active-mode-locking laser is linearly proportional to the total lasing cavity length. To implement multiple resonators having multiple reflection points installed in a sensing fiber, two types of partial reflectors are implemented for an in-line configuration, one with fiber Bragg grating and the other with a fiber Fabry–Perot interferometer. The idea behind this is interesting. However, I still have quite a number of concerns in this manuscript. There are times where there are not enough data to support the conclusions of the author. Please see some of the major concerns below.

1.The information for the sensor setup is not enough. The authors should give much more information about this. So, the readers can get its reproducibility. 

The information which is needed:

a.       A nore deep description of figure 1.

b.      What is the coupling ratio of the coupler are it is 70/30 ?

c.       What is the dc voltage range?

2.  The authors should give much more information about the novelty of this paper, especially the effect of using this new setup as sensor, which applications can be used this device?

Also, authors should replace the word “novel” to “new in the abstract section.

Are the novley is:’ the measurable length range of the sensing area is determined by the numbers 79 of PRs in the cavity’.

3. More references need to be included in the introduction part about fibers and applications using F-P.

a.“An Eight-Channel C-Band Demux Based on Multicore Photonic Crystal Fiber”, nanomaterials, (2018)

b."Super-resolvd Raman Spectra of      Toluene and Toluene-Chlorobenzene Mixture", Spectroscopy Letters, 48(6), (2015) (431-435).

c. "Superresolved Raman Spectroscopy", Spectroscopy, Lett, (2013).

4.  Much more discussion about the results should be given in this paper, especially the author needs to provide enough physicals mechanism analysis about the results.

Author Response

In this paper, the authors introduced a new type of quasi-distributed sensor system is implemented using an active-mode-locking laser cavity with multiple partially reflecting segments. The modulation frequency of the active-mode-locking laser is linearly proportional to the total lasing cavity length. To implement multiple resonators having multiple reflection points installed in a sensing fiber, two types of partial reflectors are implemented for an in-line configuration, one with fiber Bragg grating and the other with a fiber Fabry–Perot interferometer. The idea behind this is interesting. However, I still have quite a number of concerns in this manuscript. There are times where there are not enough data to support the conclusions of the author. Please see some of the major concerns below.

Point 1: The information for the sensor setup is not enough. The authors should give much more information about this. So, the readers can get its reproducibility.

Response 1: Thank you for your kind comment on not enough information on configuring sensor. We added information about the role of the parts in the set-up description.

Line 127-131 : Light reflected from long optical fiber parts, including cascaded segment sensors through a 3-port circulator, enters the ring cavity consisting of 70% of the 70/30 coupler and the semiconductor optical amplifier (SOA). The 30% port on the 70/30 coupler is used as the laser output port in Figure 1. Stage A and Stage B in Figure 1 change the segment length by applying strain between the PRs.

Point 2: The authors should give much more information about the novelty of this paper, especially the effect of using this new setup as sensor, which applications can be used this device?

Also, authors should replace the word “novel” to “new in the abstract section.

Response 2: Thank you for your kind comments on changing the word "novel" to "new". We modified the word "Novel" in this paper to "New". We also added the specific application areas of this sensor device.

Line 15-16 : A new type of quasi-distributed sensor system is implemented using an active mode locking (AML) laser cavity with multiple partially reflecting segments.

Line 64-66 : In this study, we propose a new type of quasi-distributed sensor system to interrogate the enhanced back-reflected signals from cascaded multiple partially reflecting segment sensors in the active-mode-locking (AML) laser cavity configuration.

Line 403-406 : The quasi-distributed sensor system proposed in this study will be useful various applications, such as structure health monitoring and multiple location strain sensing, to combine the advantages of a high sensitivity of the conventional point sensor and wide measurement range of the distributed sensor.

Point 3: More references need to be included in the introduction part about fibers and applications using F-P.

An Eight-Channel C-Band Demux Based on Multicore Photonic Crystal Fiber”, nanomaterials, (2018)

"Super-resolvd Raman Spectra of Toluene and Toluene-Chlorobenzene Mixture", Spectroscopy Letters, 48(6), (2015) (431-435).

"Superresolved Raman Spectroscopy", Spectroscopy, Lett, (2013).

Response 3: Thank you for your kind comments on add reference and recommend articles. We have added all of your recommended papers as a reference number 15, 16 and 17.

Line 39-42 : Interferometer sensing heads can also measure interference pattern changes due to external environment changes using an interference structure, such as a Fabry–Perot interferometer (FPI), Michelson interferometer, and Mach–Zehnder interferometer [10-17].

15.        Malka, D.; Katz, G. An Eight-Channel C-Band Demux Based on Multicore Photonic Crystal Fiber. Nanomaterials. 2018, 8, 845. https://doi.org/10.3390/nano8100845.

16.        Malka, D.; Berkovic, G.; Tischler, Y; Zalevsky, Z. Super-Resolved Raman Spectra of Toluene and Toluene–Chlorobenzene Mixture. Spectroscopy Letters. 2015, 48, 431-435. https://doi.org/10.1080/00387010.2014.905960.

17.        Malka, D.; Berkovic, G.; Hammer, Y.; Zalevsky, Z. (2013). Super-resolved Raman spectroscopy. Spectroscopy Letters. 2013, 46, 307-313. https://doi.org/10.1080/00387010.2012.728553.

Point 4: Much more discussion about the results should be given in this paper, especially the author needs to provide enough physicals mechanism analysis about the results.

Response 4: Thank you for your kind comments on the lack of discussion on the results. We added information the stability, minimum measurable length change and limit of the PR number for more physical mechanism analysis.

Line 334-342 : Figures 12 show the length measurement stability of PR1 when the position of PR1 did not change for 160 min and the stability of the sensor system was measured to have a standard deviation of about 10.2 μm, which corresponds to the minimum measurable length change.

In this research, the sensing positions were experimentally implemented using three FBGs of PRs. To calculate the limit of sensing point numbers, the intensity reduction of each additional lasing peak is monitored and it is measured to about 0.3 dB per added FBG-PR as shown in Figure 7 (b) and Figure 10 (b). It means that more than 80 FBG-PR can be cascaded in the single sensing port for multiple reflection points assuming the undistinguished SNR level of multiple lasing spectra of 48 dB.

Reviewer 3 Report

Quasi-distributed active-mode-locking laser interrogation with multiple partially reflecting segment

The authors present a quasi-distributed sensor system which is implemented using an active-mode-locking laser cavity with mutiple partially reflecting segments (PRs). By measuring the modulation frequency, the strain applied to fiber segments between PRs is obtainded.  The authors believe that these are relevant mentioning that the back-reflected signal intensity of the PRs is important and that the number of these in the sensor arrangment is determined by their reflectivities. But, it is not clear the importance of the PRs characteristics for the measurement of the change in length. Therefore, I sugest to reject the article. 

However, I have the following comments and suggestions for the authors:

1.     I suggest to mention some relevant advantages of point-based sensors, as it is mentioned in line 37.

2.     In lines 46-47, the authors write that the complex fabrication process and the high cost are disadvantages of discrete sensing heads. But this is not entirely true, since there exist interferometric sensors that are very easy to fabricate by using, for example, single mode fibers and a fusion splicer. So, I suggest to rewrite this information and to add the corresponding references.

3.      It is not clear which the authors want to say in lines 74-76.

4.     I suggest to add a reference about the number of sensing points limit in a quasi-distributed FBG sensor (line 71).

5.     In the introduction, the authors basically mention that the back-reflected signal intensity is important and that the number of PRs is determined by their reflectivities, but in the manuscript (section 2) this issue is not discussed.  

6.     I suggest to rewrite the section 2. For clarity puroposes, I reccomment first to write about the experimental setup, then about the operation principle.

-       The coupler is not used only as a output port

-       The stage A and the stage B are not mentioned in the second paragraph

-       In the first paragraph of the section is not clear as the change in length as a function of the frequency is obtained.

Apparently, the PRs are used only to generate multiple laser cavities, because it is not explained why the reflectance of the PRs is crucial in the measurement of the change in length as a function of the frequency. Besides, it is not explained how the  reflectance value limits the number of PRs for the design of a quasi-distributed sensor, etc. Therefore, I  do not find the relevance of adding a section about the PRs characterization.

Now if the authors explaine such relevance, I have the following comments and suggestions:

-       I suggest to add a reference in line 150 (about conventional FBG interrogation method based on…)

-       The title of the section is “Characterization of the PR”, then I suggest to add a figure showing the reflection spectrum of the FBG used in their exeriments.

-       Please write a number (line 163), instead to write “the resulting small transmisión loss”

-       I think,  all the equations are not necessary; I recommend to show and to explain only equation 5.   

-       Why the FPIs  were characterized at 1310 nm wavelength? It is necessary to explain it.

-       In the manuscript, it is not mentioned what is DS.

-       In Figure 3, It is better to show the reflection spectrum because the interference pattern is not clear.

7.     I suggest to add a subsection about the sensor characteristics, such as: sensitivity, minimun detectable length change, rango, repeatibility..

8.     I suggest to show the experimental change in length as a function of the modulation frequency of all the PRs in each subsection. Besides, I also recommend to show in the same figures the applied strain because this is mentioned in the manuscript.

9.     I recommend to show only the laser emission spectra when these vary due to the applied change in length. Here, why are their spectra partially showed? I think it is better to show all the laser emission spectra.  

10.  The Fig. 8 is the same that the Fig. 10, these should be different.  

11.  Why the cavity length mentioned in line 270 is not the same  that the one mentioned in line 239?. It is supposed that the cavity length was not modified. Please, explaine it.

12.  How were the distance linewidths and  the minimun distance between PR segments obtained?

13.  Please revise the information written in lines 258-260 , 271,  and 292-294

Author Response

The authors present a quasi-distributed sensor system which is implemented using an active-mode-locking laser cavity with mutiple partially reflecting segments (PRs). By measuring the modulation frequency, the strain applied to fiber segments between PRs is obtainded.  The authors believe that these are relevant mentioning that the back-reflected signal intensity of the PRs is important and that the number of these in the sensor arrangement is determined by their reflectivities. But, it is not clear the importance of the PRs characteristics for the measurement of the change in length. Therefore, I suggest to reject the article.

However, I have the following comments and suggestions for the authors:

Point 1: I suggest to mention some relevant advantages of point-based sensors, as it is mentioned in line 37.

Response 1: Thank you for your kind comments on adding the relevant advantages of point-based sensors. We agreed the advantages of the point sensor were not written well and thus added the advantages of the point sensor in introduction and more reference.

Line 42-45 : The small length of point-based sensing heads using an FBG sensor or interferometer sensor offers a high position resolution along optical fiber in addition to the advantages of high sensitivity, geometric diversity, and rapid response [18]. The discrete sensing position along the optical fiber is determined by the separated locations of multiple sensing heads.

Point 2: In lines 46-47, the authors write that the complex fabrication process and the high cost are disadvantages of discrete sensing heads. But this is not entirely true, since there exist interferometric sensors that are very easy to fabricate by using, for example, single mode fibers and a fusion splicer. So, I suggest to rewrite this information and to add the corresponding references.

Response 2: Thank you for your kind comments on rewriting this information about the complex manufacturing process. The sentence was revised to clarify this meaning and the corresponding reference was added.

Line 47-50 : The point-based sensors suffer from an extra cost of each sensing heads and limited number of discrete sensing positions. Fabrication process of sensing heads can be simplified when sensing components are based on the optical fiber [11].

Point 3: It is not clear which the authors want to say in lines 74-76.

Line 74-76 : Since the laser is oscillated only when the modulation frequencies for the mode locking condition match with the corresponding resonator lengths, it is possible to read the multiple partially reflecting segments along the sending fiber.

Response 3: Thank you for your kind comments on the clarity of the line 74-76. We modified the sentence to clarify that the modulation frequency of the active mode-locked laser differs according to the cavity length and that it can be used to search locate the PR.

Line 77-81 : The AML laser oscillates at the specific modulation frequency that matches to the mode locking frequency of cavity length. It means there can be multiple mode locking frequencies corresponding to the multiple cavity lengths, respectively. Thus, it is possible to read the lengths of multiple partially reflecting segments along the sensing fiber by using the mode locking frequency of the AML laser [25].

Point 4: I suggest to add a reference about the number of sensing points limit in a quasi-distributed FBG sensor (line 71).

Response 4: Thank you for your comment on adding a reference to the number of detection points limit in quasi-distributed FBG sensors. We added ref 1 and ref 9, and modified the sentence to explain what limits the number of quasi-distributed FBG sensors.

Line 70-75 : For example, the available sensing point number of FBG in a single fiber is limited below 20 from the ratio between the total wavelength band of light source and individual wavelength band of each FBG [1-9]. Instead, the controllable split ratio between back-reflected and transmitted signal intensities is more important parameter of PR for the sensing point number limit of proposed reflectometry because the limit is determined from the intensity reduction of each additional lasing peak.

Point 5: In the introduction, the authors basically mention that the back-reflected signal intensity is important and that the number of PRs is determined by their reflectivities, but in the manuscript (section 2) this issue is not discussed.

Response 5: Thank you for your comment on adding a discussion on PR numbers. The number of PRs is affected by the intensity reduction of the multiple lasing spectra. We added a sentence to the conclusion to explain how the number of PRs is limited by the results of experiments using added FBGs.

Line 337-342 : In this research, the sensing positions were experimentally implemented using three FBGs of PRs. To calculate the limit of sensing point numbers, the intensity reduction of each additional lasing peak is monitored and it is measured to about 0.3 dB per added FBG-PR as shown in Figure 7 (b) and Figure 10 (b). It means that more than 80 FBG-PR can be cascaded in the single sensing port for multiple reflection points assuming the undistinguished SNR level of multiple lasing spectra of 48 dB.

Point 6: I suggest to rewrite the section 2. For clarity puroposes, I reccomment first to write about the experimental setup, then about the operation principle.

-       The coupler is not used only as a output port

-       The stage A and the stage B are not mentioned in the second paragraph

-       In the first paragraph of the section is not clear as the change in length as a function of the frequency is obtained.

Response 6: Thank you for your comment regarding the rewriting of Section 2. We added a description of the role of the coupler and stage in section 2 and modified the content of the sentence and the placement of the sentence to clarify the relationship between modulation frequency and length.

Line 127-130 : . Light reflected from long optical fiber parts, including cascaded segment sensors through a 3-port circulator, enters the ring cavity consisting of 70% of the 70/30 coupler and the semiconductor optical amplifier (SOA). The 30% port on the 70/30 coupler is used as the laser output port in Figure 1.

Line 96-103 : As an external modulation frequency applies to the gain element, the AML peak laser output is generated only when the round trip time of light passing through the laser cavity exactly matches (or an integer multiple of) the period of the modulated frequency. At this time, the length change of laser cavity affects the round trip time of light and thus the mode locking frequency is also affected from the change of round trip time of light. This means that the cavity length can be directly detected by measuring the intensity variation of AML laser output. Therefore, the changes in cavity length due to external changes, such as temperature and deformation, can be observed by scanning the modulation frequency.

Point 7: Apparently, the PRs are used only to generate multiple laser cavities, because it is not explained why the reflectance of the PRs is crucial in the measurement of the change in length as a function of the frequency. Besides, it is not explained how the  reflectance value limits the number of PRs for the design of a quasi-distributed sensor, etc. Therefore, I do not find the relevance of adding a section about the PRs characterization.

Now if the authors explaine such relevance, I have the following comments and suggestions:

7-1: -     I suggest to add a reference in line 150 (about conventional FBG interrogation method based on…)

7-2: -       The title of the section is “Characterization of the PR”, then I suggest to add a figure showing the reflection spectrum of the FBG used in their exeriments.

7-3: -       Please write a number (line 163), instead to write “the resulting small transmisión loss”

7-4: -       I think, all the equations are not necessary; I recommend to show and to explain only equation 5.  

7-5: -       Why the FPIs were characterized at 1310 nm wavelength? It is necessary to explain it.

7-6: -       In the manuscript, it is not mentioned what is DS.

7-7: -       In Figure 3, It is better to show the reflection spectrum because the interference pattern is not clear.

Response 7: Thank you for your delicate and kind comments on the clarity and detail of this paper.

7-1: A reference related to the conventional FBG interrogation method was added.

Line 156-159 : For cascaded multiple FBGs as a point-based sensor system, the conventional FBG interrogation methods based on the wavelength-domain information required a strong, narrow, and non-overlapped reflection spectrum of the FBG [9]..

9.      Peng, P. C.; Lin, J. H.; Tseng, H. Y.; Chi, S. Intensity and wavelength-division multiplexing FBG sensor system using a tunable multiport fiber ring laser. IEEE Photonics Technology Letters. 2004, 16, 230-232. https://doi.org/10.1109/LPT.2003.818916.

7-2: Experimental result of reflected light and transmitted light spectra of FBG was added.

Figure 2. Reflection and transmission spectra of FBGs.

7-3 : FBG transmission loss was added.

Line 174-176 : As shown in Figure 2 with the reflection and transmission spectra, the transmission loss of the FBG used in the experiment was measured to be about 0.3 dB.

7-4 : The equation (2-4) was deleted and only equation (5) was left.

Line 210

7-5 : Since the center wavelength of the SOA light source used in the measurement experiment for FBG was 1310 nm, the spectral measurement and characterization for FPI are also implemented around 1310 nm region. We added this explanation to the figure description.

Figure 6. Experimental measurement of intensity variation of reflected and transmitted light according to ΔS of optical fiber FPI using the amplified spontaneous emission (ASE) of SOA with the center wavelength of 1310 nm.

7-6 : Explanation of DS in the manuscript.

Line 193-194 : ΔS is the spacing between the two cutting surfaces, and the cutting surface is fixed to the tube at various ΔS.

7-7 : Adding FPI Reflection Spectrum

Figure 5. (a - e) FPI reflection spectrum according to ΔS, (f) Relationship between ΔS and wavenumber variation Δk.

Point 8: I suggest to add a subsection about the sensor characteristics, such as: sensitivity, minimun detectable length change, range, repeatibility.

Response 8: Thank you for your kind comments on adding subsection about the sensor characteristics, such as: sensitivity, minimum detectable length change, range, repeatibility. We have added a graph and description of information on sensitivity, repeatability, and so on.

Figure 9. Length measurement stability results of (a) PR1 and (b) PR2

Line 285-290 : Figures 9(a) and (b) show the length measured stability of PR1 and PR2. Since the positions of PR1 and PR2 and the cavity length between them did not change during the measurement experiment for 160 minutes, the results in Figure 9 show the stability of the sensor system and the corresponding minimum measurable length. The standard deviation of PR1 was measured to about 4.4 μm, and that of PR2 was about 4.5 μm. It means that the minimum measurable length change of the sensor system is limited to about 4.5 μm.

Figure 12. Length measurement stability results of PR1

Line 334-336 : Figures 12 show the length measurement stability of PR1 when the position of PR1 did not change for 160 min and the stability of the sensor system was measured to have a standard deviation of about 10.2 μm, which corresponds to the minimum measurable length change.

Figure 15. Length measurement stability results of (a) PR1 and (b) PR2

Line 388-392 : Figures 15 (a) and (b) show the length measured stability of PR1 and PR2. Since position of PR1 and PR2 did not change for 120 min, the results in Figure 15 show the stability of the sensor system and the corresponding minimum measurable length. The standard deviations of PR1 and PR2 were about 10.6 μm and 9.2 μm, respectively. Thus, the minimum measurable length change of the sensor system is limited to about 10.6 μm.

Point 9: I suggest to show the experimental change in length as a function of the modulation frequency of all the PRs in each subsection. Besides, I also recommend to show in the same figures the applied strain because this is mentioned in the manuscript.

Response 9: Thank you for your kind comments on the relation between length change and modulation frequency. As you suggested, we included those relations for each PRs in Figure 8, 11, and 14.

Figure 8. Linear relationship between the mode locking frequency of PR3 and the change in length of the PR2–PR3 segment (stage B in Figure 1).

Figure 11. Linear relationships between the mode locking frequencies of (a) PR2 and (b) PR3 and the change in length of the PR1–PR2 segment (stage A in Figure 1).

Figure 14. Relationships between the mode locking frequencies of (a) PR3 and (b) PR4 and the change in length of the PR2–PR3 segment (stage B in Figure 1).

For the relations between applied strain and length change for each PRs in Figure 8, 11, and 14, both of applied strain and length change are proportional to the change of modulation frequency. However, in our sensing system, it is recommended to show the unit of length change, instead of applied strain, because the applied strain is also dependent to the distance between fiber holders in the center region of the segment. Therefore, Figure 8, 11, and 14 are presented with the unit of length change and the calculated values of applied strains and linearities are mentioned in the corresponding manuscripts like below.

Line 283-284 : The sensitivity of the PR3 mode locking-frequency shift with respect to the PR2–PR3 segment length increment is 594 Hz/mm, with a linearity R2 value of 0.9946.

Line 322-324 : The sensitivity of the PR2 mode locking-frequency shift to the PR1–PR2 segment length increment is 461 Hz/mm; the sensitivity with respect to PR3 is 594 Hz/mm. The R2 values for both graphs are larger than 0.999.

Line 381-383 : The strain sensitivities of the PR3 and PR4 mode locking frequencies were 330 Hz/mm and 315 Hz/mm, while the R2 values corresponding to very linear relationships were 0.9955 and 0.9957, respectively.

Point 10: I recommend to show only the laser emission spectra when these vary due to the applied change in length. Here, why are their spectra partially showed? I think it is better to show all the laser emission spectra.

Response 10: Thank you for your kind comments on the partially expressed spectrum. It is also available to show the all the lasing spectra in a single window, but we found that the partial presentation of top intensity height part is the best to distinguish the peak change clearly for all of PRs because the width of each PRs’ peak is not same. The other graph lines of middle and lower height part do not show any meaning of spectral shift but looks harmful to catch the peak change. I hope to you understand the presentation skill when it includes scientific results.

Point 11: I The Fig. 8 is the same that the Fig. 10, these should be different.

Response 11: Thank you for your kind comment on the fact that Figure 8 and Figure 10 are the same. It was our mistake and we update Figure 11 with the correction.

Figure 11. Linear relationships between the mode locking frequencies of (a) PR2 and (b) PR3 and the change in length of the PR1–PR2 segment (stage A in Figure 1).

Point 12: Why the cavity length mentioned in line 270 is not the same that the one mentioned in line 239?. It is supposed that the cavity length was not modified. Please, explaine it.

Response 12: Thank you for your kind comment on the different resonator lengths for each experiment. To characterize the sensor at various modulation frequencies intentionally, we have set each experiment to have a different PR position and fiber length so that the modulation frequency peaks are set differently.

Point 13: How were the distance linewidths and the minimum distance between PR segments obtained?

Response 13: Thank you for your kind comments on calculating the PR segment minimum distance. The sensor system of this paper measures the change of the cavity length by measuring the movement of the modulation frequency peak. Therefore, each distance linewidth is limited by the linewidth of the modulation frequency peak. The minimum distance between PR segments is determined not to overlap each other peaks. We added a description of the minimum distance to the paper.

Line 341-345 The minimum distance of the PR segment can be determined based on the distance linewidth from the modulation frequency peak since the modulation frequency peaks of the PR segments must be distinguished without overlapping each other. In addition, since the sensor system measures the change in the length of the PR segment, the minimum distance of the PR segment can be used to determine the spatial position resolution along optical fiber.

Point 14: Please revise the information written in lines 258-260 , 271, and 292-294

Response 14: Thank you for your kind comments on the need to revise the picture number. This was our mistake and we continued to correct it.

Line 273-276 : Figures 7(d) and (e) show the modulation frequency spectrum of PR1 and PR2, respectively, which did not change within a negligible measurement error as the length of the PR1–PR2 segment did not change. Figure 7(f) shows the spectral change of PR3 along modulation frequency domain when the segment length increases from 0 to 1.4 mm.

Line 315-318 : Figure 10(d) shows the change of the PR1 mode locking frequency, revealing that no considerable change in the mode locking frequency was observed, with a negligible measurement error. Figures 10(e) and 10(f) show the shifts of the mode locking frequencies upon the total cavity length changes of PR2 and PR3, respectively.

Line 371-374 : Figures 13(d) and (e) show that the mode locking frequencies of PR1 and PR2 are unchanged, i.e., unaffected by the change of the PR2–PR3 segment length. Figures 13(f) and (g) show the variations of the mode locking frequencies of the PR3 and PR4, installed the far location after the PR2–PR3 segment, respectively.

Round  2

Reviewer 3 Report

 The authors have almost satisfactorily responded most of my comments and suggestions.

The points from 8-10 were not totally taken into account and were not satisfactorily answered. For example, in point 8, the way they calculate  the minimum detectable length change is wrong. This is estimated as the change in length for which the signal to noise ratio is equal to 1.

Author Response

Response to Reviewer 3 Comments

The authors have almost satisfactorily responded most of my comments and suggestions.

The points from 8-10 were not totally taken into account and were not satisfactorily answered. For example, in point 8, the way they calculate  the minimum detectable length change is wrong. This is estimated as the change in length for which the signal to noise ratio is equal to 1.

Point 8: I suggest to add a subsection about the sensor characteristics, such as: sensitivity, minimun detectable length change, range, repeatibility.

Response 8: Thank you for your kind comments on adding subsection about the sensor characteristics, such as: sensitivity, minimum detectable length change, range, repeatability. The descriptions on the sensitivity and range are written in the lines 283 ~ 286,  lines 327 ~ 331, lines 391 ~ 395 in detail.

Based on the new reference of [26] on the relations among the standard deviation, limit of detection (LOD) and limit of quantification (LOQ), a signal-to-noise ratio (SNR) of three is generally accepted for estimating LOD and signal-to-noise ratio of ten is used for estimating LOQ. Thus, we additionally described on the minimum detectable length change from the standard deviation of repeatability on Figure 9, 12, and 15 in detail.

[26] Shrivastava, A.; Gupta, V. B. Methods for the determination of limit of detection and limit of quantitation of the analytical methods. Chronicles of Young Scientists. 2011, 2, 21. https://doi.org/10.4103/2229-5186.79345.                                             

Figure 9. Length measurement stability results of (a) PR1 and (b) PR2

Line 287-295 : Figures 9(a) and (b) show the length measured stability of PR1 and PR2. Since the positions of PR1 and PR2 and the cavity length between them did not change during the measurement experiment for 160 minutes, the results in Figure 9 show the stability of the sensor system and the corresponding minimum measurable length. The standard deviation of PR1 was measured to about 4.4 μm, and that of PR2 was about 4.5 μm. It means that the minimum measurable length change of the sensor system is limited to about 4.5 μm. Since a SNR of three is generally accepted for estimating limit of detection (LOD) and SNR of ten is used for estimating and limit of quantification (LOQ) [26], it can be expected that the minimum measurable length change of the sensor system can be limited to more than about 45 μm.

Figure 12. Length measurement stability results of PR1

Line 341-344 : Figures 12 show the length measurement stability of PR1 when the position of PR1 did not change for 160 min and the stability of the sensor system was measured to have a standard deviation of about 10.2 μm, which corresponds to the minimum measurable length change of more than about 102 μm from the LOQ of the ten times of standard deviation.

Figure 15. Length measurement stability results of (a) PR1 and (b) PR2

Line 398-402 : Figures 15 (a) and (b) show the length measured stability of PR1 and PR2. Since position of PR1 and PR2 did not change for 120 min, the results in Figure 15 show the stability of the sensor system and the corresponding minimum measurable length. The standard deviations of PR1 and PR2 were about 10.6 μm and 9.2 μm, respectively. Thus, the minimum measurable length change of the sensor system is estimated to more than about 106 μm based on the LOQ with the SNR of ten.

Point 9: I suggest to show the experimental change in length as a function of the modulation frequency of all the PRs in each subsection. Besides, I also recommend to show in the same figures the applied strain because this is mentioned in the manuscript.

Response 9: Thank you for your kind comments on the relation between length change and mode locking frequency. As you suggested, we included those relations between length change and mode locking frequency for each PRs in Figure 8, 11, and 14.

For the relations between applied strain and length change for each PRs in Figure 8, 11, and 14, both of applied strain and length change are proportional to the change of modulation frequency. However, in our sensing system, it is recommended to show the unit of length change, instead of applied strain, because the applied strain is also dependent to the distance between fiber holders in the center region of the segment. Therefore, Figure 8, 11, and 14 are presented with the unit of length change, but the calculated values of applied strains and linearities are additionally described in the corresponding manuscripts like below.

In addition, the comparisons between the mode locking frequency/ length change and mode locking frequency/applied strain are added for the easy understand these relations.

 Line 283-286 : The sensitivity of the PR3 mode locking-frequency shift with respect to the PR2–PR3 segment length increment is 594 Hz/mm, with a linearity R2 value of 0.9946. Based on the distance between fiber holders in the center region of the segment of 0.75 m, this sensitivity of length change is corresponding to the sensitivity of applied strain with 445.5 Hz/mϵ.

 Line 327-331 : The sensitivity of the PR2 mode locking-frequency shift to the PR1–PR2 segment length increment is 461 Hz/mm; the sensitivity with respect to PR3 is 594 Hz/mm. The R2 values for both graphs are larger than 0.999. Based on the distance between fiber holders in the center region of the segment of 0.75 m, this sensitivity of length change is corresponding to the sensitivity of applied strain with 345.75 Hz/mϵ.

  Line 391-395 : The strain sensitivities of the PR3 and PR4 mode locking frequencies were 330 Hz/mm and 315 Hz/mm, while the R2 values corresponding to very linear relationships were 0.9955 and 0.9957, respectively. Based on the distance between fiber holders in the center region of the segment of 0.55 m, this sensitivity of length change is corresponding to the sensitivity of applied strain with 173.25 Hz/mϵ.

 Point 10: I recommend to show only the laser emission spectra when these vary due to the applied change in length. Here, why are their spectra partially showed? I think it is better to show all the laser emission spectra.

 Response 10: Thank you for your kind comments on the partially expressed spectrum. It is also available to show the all the lasing spectra in a single window, but we found that the partial presentation of top intensity height part is more suitable to distinguish the peak change clearly for all of PRs. The other graph lines of middle and lower height parts do not show any meaning of spectral shift but looks harmful to focus on the peak change only of laser emission spectra. I hope the reviewer to understand the presentation format to deliver our main intantion of experimental results.
